# Does Insulin Treatment Affect Umbilical Artery Doppler Indices in Pregnancies Complicated by Gestational Diabetes?

**DOI:** 10.3390/healthcare12191972

**Published:** 2024-10-03

**Authors:** Libera Troìa, Stefania Ferrari, Anna Dotta, Sonia Giacomini, Erika Mainolfi, Federica Spissu, Alessia Tivano, Alessandro Libretti, Daniela Surico, Valentino Remorgida

**Affiliations:** 1Department of Gynaecology and Obstetrics, University Hospital Maggiore della Carità, 28100 Novara, Italy; stefania.ferrari1994@gmail.com (S.F.); annadotta94@gmail.com (A.D.); giacomini.sonia@yahoo.it (S.G.); erika.mainolfi81@gmail.com (E.M.); federica.spissu@outlook.it (F.S.); alessia.tivano@gmail.com (A.T.); libretti.a@gmail.com (A.L.); daniela.surico@med.uniupo.it (D.S.); valentino.remorgida@uniupo.it (V.R.); 2Department of Translational Medicine, University of Eastern Piedmont, 28100 Novara, Italy

**Keywords:** gestational diabetes mellitus, maternal hyperglycemia, insulin therapy, nutritional therapy, umbilical artery flow, fetal circulation

## Abstract

**Background/Objectives:** Gestational diabetes mellitus (GDM) is one of the most common morbidities of pregnancy. The impact of increased maternal blood glucose on fetoplacental hemodynamics is not fully elucidated, especially in patients with uncontrolled GDM necessitating insulin therapy. The objective of this study was to assess the impact of insulin therapy on the umbilical artery dopplers in GDM pregnancies adequate for gestational-age fetuses. **Methods:** Retrospective observational study among 447 GDM pregnant women, divided according to their treatment (nutritional therapy (NT), long acting (LA) insulin, combined insulin) and 100 healthy controls with the same gestational age. The umbilical artery pulsatility index (UA-PI) was recorded at 28, 32 and 36 weeks. **Results:** UA-PI values declined in both GDM and healthy controls at all three time intervals. The combined insulin group showed reduced UA-PI values in comparison to the LA insulin group, but the difference never reached statistical significance. The combined insulin group exhibited significantly reduced UA-PI values at 32- and 36-weeks’ gestation compared to the NT groups. **Conclusions:** A decreased impedance to blood flow in the umbilical artery of diabetic mothers on insulin therapy was observed. This was more pronounced during the last trimester. The extent to which umbilical artery PI can predict unfavorable outcomes has yet to be determined. Further additional studies are necessary to confirm the precise impact of glucose levels and medical interventions on the circulation of both the fetus and the mother.

## 1. Introduction

Gestational diabetes mellitus (GDM) is one of the most common morbidities of pregnancy. The incidence of GDM has increased over the past 30 years due to a worldwide rise in overweight and obesity rates, as well as an increase in the number of pregnant women over the age of 35 [1].

Hyperglycemia poses a substantial danger for negative outcomes in both mothers and newborns [1]. This encompasses elevated occurrences of fetal macrosomia, fetal acidosis, neonatal hypoglycemia, admission to the neonatal intensive care unit (NICU), and cesarean birth due to fetal distress [2]. Finally, GDM long-term outcomes on both mother and offspring are concerning [2].

Perinatal therapy in GDM pregnant women include metabolic management, rigorous surveillance, and serum glucose level-monitoring [1]. Nutritional therapy remains the first-line treatment; however, suboptimal glycemic blood levels require medications such as metformin, glibenclamide (glyburide), and/or insulin [1,3].

Maternal diabetes may alter placental circulation, thus influencing endothelial function [4]. Doppler velocimetry is a tool used for perinatal surveillance, estimating the level of vascular resistance in placental circulation. Blood flow measurement in the fetal umbilical (UA) and middle cerebral artery (MCA) is a reliable and noninvasive method for evaluating fetal hemodynamics. This technique is particularly useful in pregnancies that are affected by intrauterine growth restriction (IUGR) or preeclampsia [5].

The correlation between diabetes and fetal doppler velocimetry has been previously studied, with conflicting data [6,7]. UA and MCA fetal doppler velocimetry was not found altered for some authors [8,9], while others have found significant hemodynamic changes in GDM patients [10,11] and also unfavorable neonatal outcomes [12,13]. Nevertheless, GDM poses an important socio-economical challenge in the world [1], and its prevalence in Italy is attested to be around 11% [11]. In their systematic review, Rane et al. suggested that an abnormal umbilical artery pulsatility index (UA-PI) may be more clinically valuable in predicting worse neonatal outcomes in diabetes-complicated pregnancies compared to the cerebroplacental ratio and middle cerebral artery pulsatility index [14].

Also, the impact of increased maternal blood glucose on fetoplacental hemodynamics is far from being elucidated [9,15], especially in patients with uncontrolled GDM necessitating insulin therapy. Considering the UA PI as a marker of fetal well-being and knowing that GDM could affect the maternal–fetal circulation, the objective of this study was thus to assess the impact of insulin therapy on the umbilical artery dopplers in GDM pregnancies with adequate-for-gestational-age (AGA) fetuses.

## 2. Materials and Methods

### 2.1. Study Design

A retrospective observational study was conducted at the Maternal Fetal Unit of the Department of Gynecology and Obstetrics, University Hospital Maggiore della Carità in Novara, Piedmont, Italy. The study population included 447 GDM pregnant women from January 2020 to March 2024. This agreed with the protocols of the International Declaration of Helsinki.

### 2.2. Study Setting

A 75 g oral glucose tolerance test (OGTT) was used as screening test for GDM. According to Italian national guidelines in high-risk women for GDM (previous GDM, body mass index (BMI) greater than 30 kg/m^2^, fasting plasma glucose levels between 100 and 125 mg/dL according to the first trimester blood test [16]), the test was conducted at 16–18 weeks and if negative repeated between 24 and 28 weeks [16]. In low-risk women (≥35 years, BMI between 25 and 30 kg/m^2^, family history of diabetes, previous macrosomia newborn (>4000 g), high risk for diabetes ethnicity) the test was performed directly between 24 and 28 weeks of gestation [16].

Following diagnosis, patients were instructed on self-monitoring their blood glucose levels. Tailored dietary guidance and lifestyle changes were given to manage weight gain and maintain a stable blood glucose level during pregnancy. Periodic visits for monitoring compliance to therapy (either nutritional or pharmacological) were individualized; on those occasions, glycemic values and type of treatment were recorded until delivery.

If over 50% of blood glucose measurements exceeded the target range during the initial 2-week assessment, insulin therapy was indicated, in accordance with the Italian National Guidelines [16]. A long-acting insulin analogue administered after the evening meal was the starting therapy, but if the postprandial values were also still out of range a rapid insulin analogue was added on demand.

### 2.3. Study Population, Inclusion Criteria and Control Group

The women were categorized into three groups based on their treatment, as follows: group 1—nutritional therapy (NT) only (n = 287); group 2—long-acting (LA) insulin analogue (Glargine) (n = 97); group 3—combined long- and rapid-acting (Lispro) insulin analogue (combined therapy) (n = 63). The most recent therapy reported was considered when categorizing each patient based on the treatment regimen they followed.

A randomly selected group of 100 healthy pregnant women matched in terms of maternal age and BMI served as the control group.

We routinely collect maternal characteristics (including age, gestational weight gain (GWG), BMI before pregnancy and at delivery, ethnicity, smoking status, family history of diabetes, and pregestational maternal diseases), obstetric history (mode of conception, parity, and previous or current pregnancy complications), as well as delivery and neonatal outcomes (including gestational age at delivery, mode of delivery, APGAR score at 1st and 5th min, NICU admission, sex, and birthweight) for each patient attending. The study also assessed the values of OGTT and the management of GDM using either NT alone or NT combined with insulin, as well as the type of insulin analogue used, until delivery.

These data along with ultrasound measurements (Samsung Healthcare, Seoul, Republic of Korea), including Doppler assessment, are stored on an electronic database (Astraia software, version 1.27.1, GmbH, Munich, Germany). Institutional Review Board approval was obtained with the protocol number Prot. 753/CE.

### 2.4. Exclusion Criteria

The exclusion criteria for this study included the diagnosis of either Type 1 or Type 2 Diabetes mellitus, the detection of fetal abnormality during ultrasound, the presence of coexisting conditions such as hypertensive disorders or abnormal fetal growth (including fetal growth restriction, being small for gestational age, or being large for gestational age (LGA)), and incomplete follow-up.

### 2.5. Outcome Measures and Methods

The ultrasound examinations and umbilical artery (UA) velocity waveform recordings were conducted using the Samsung HERA W9 and Samsung WS80 ultrasound equipment (Samsung Healthcare, Seoul, Republic of Korea). All the examinations were performed with transabdominal probes operating at frequencies of 2–5 MHz, using color and pulsed wave Doppler features. To minimize inter-observer variability, all ultrasounds were conducted by a skilled operator certified by The Fetal Medicine Foundation (FMF) (https://fetalmedicine.org/lists/map/certified/fetal-doppler-ultrasound, accessed on 21 September 2024).

UA Doppler measurements were acquired on a median free loop of the umbilical cord. UA-PI was assessed by measuring it over a minimum of three consecutive, consistent cardiac cycles. The Doppler measurements were conducted while the fetus was at rest, as the flow parameters of fetal Doppler might be influenced by fetal movements and the fetal heart rate (FHR) [9].

The umbilical artery pulsatility index (UA-PI) was measured at three time intervals during pregnancy in all groups—at 28 weeks (T0), 32 weeks (T1), and 36 weeks (T2). During each ultrasound examination, measurements of fetal biometrics (abdominal circumference, head circumference, biparietal diameter, femur length) and assessments of amniotic fluid (AF) were conducted. Ultrasound assessments were performed in accordance with the guidelines set by ISUOG [5].

According to State privacy law, every patient undertaking a clinical examination including ultrasound scans must give their informed consent to the collection and usage of their anonymized clinical data.

### 2.6. Statistical Analysis

As stated, all clinical and ultrasound data were recovered from the electronic database (Astraia). The mean and standard deviation were used to compute quantitative variables, while absolute and relative frequencies were used to calculate qualitative variables. The Student’s *t*-test and one-way ANOVA were employed to compare the means of the two or more groups for continuous variables that follow a normal distribution. The Mann–Whitney U test was employed for non-normally distributed continuous variables. The chi-squared test or Fisher’s exact test, if applicable, were employed to compare categorical variables. The threshold for statistical significance was established at a *p*-value of less than 0.05. The statistical studies were conducted using the Graph Pad Prism 6 software.

## 3. Results

The analysis on our cohort of 447 GDM pregnant women showed that 287 of them (64.2%) were able to control GDM using non-insulin treatments alone, whereas the remaining 160 (35.8%) needed insulin therapy.

No notable differences between the two groups in terms of age, method of conception, or smoking behavior were observed (Table 1).

In the NT group, a higher percentage of women were nulliparous (120 (41.8%) vs. 46 (28.7%), *p* = 0.0078) and Caucasian (217 (75.6%) vs. 89 (55.6%), *p* = 0.0001). Patients who required insulin therapy had higher pregestational BMI and BMI at delivery compared to those who did not (pregestational BMI = 27.78 ± 5.04 Kg/m^2^ vs. 25.46 ± 5.33 Kg/m^2^, *p* = 0.0001; BMI at delivery = 30.36 ± 4.63 Kg/m^2^ vs. 28.61 ± 5.04 Kg/m^2^; *p* value = 0.0003). On the other hand, the NT group had a higher GWG compared to the other group, with a mean of 8.44 ± 5.39 Kg versus 7.21 ± 5.60 Kg, and this difference was statistically significant (*p* = 0.0233) (Table 1).

As expected, the insulin therapy group had a higher prevalence of family history of diabetes, previous GDM, and macrosomia compared to the non-insulin therapy group. The numbers and percentages were 81 (50.6%), 40 (25%), and 16 (10%) in the insulin therapy group, and 114 (39.7%), 33 (11.5%), and 7 (2.4%) in the NT group, respectively. The differences were statistically significant, with *p*-values of 0.0288, 0.0012, and 0.0003, respectively (Table 1).

When comparing the OGTT values of the two groups, those receiving insulin therapy had higher levels of fasting plasma glucose (FPG) (103.10 ± 71.88 mg/dL vs. 89.39 ± 9.19 mg/dL, *p* = 0.0015), higher levels of glucose at 1 h (173.24 ± 30.86 mg/dL vs. 165.07 ± 30.9 mg/dL, *p* = 0.0076), and higher levels of glucose at 2 h (144.80 ± 34.59 mg/dL vs. 138.13 ± 28.21 mg/dL, *p* = 0.0278). Women who required insulin therapy had significantly higher fasting plasma glucose levels in the first trimester compared to those who did not (95.16 ± 20.47 mg/dL vs. 83.10 ± 10.55 mg/dL, *p* = 0.0001), as well as higher levels of glycated hemoglobin (5.54 ± 0.48% vs. 5.18 ± 0.31%, *p* = 0.0001) (Table 1).

No significant changes in terms of gestational age at birth, hypertensive disorders, method of delivery, or post-partum hemorrhage between women who received NT alone and those who received NT in addition to insulin were seen, as shown in Table 2. Nevertheless, women treated with insulin exhibited a notably higher rate of labor induction (63 (39.4%) vs. 74 (25.8%), *p* = 0.0038).

Regarding perinatal outcomes, no significant differences in terms of fetal sex, birth weight (NT 3148.84 ± 533.30 g vs. NT plus insulin 3248.93 ± 501.87 g, *p* = 0.0527), Apgar score, or NICU admission between the two groups were found (Table 2).

In addition, among the 160 insulin-treated patients, long-acting analogues (LA insulin group) were sufficient in 97 (60.6%) of them, with the remaining 63 (39.4%) needing a combination of long- and rapid-acting analogues (combined therapy group). As stated earlier, UA-PI values were collected at three distinct time intervals (28, 32, and 36 weeks of gestation).

The first analysis performed between all GDM women and controls did not show any difference (Table 3). As a general trend, the UA-PI values declined with the progression of pregnancy in both GDM and healthy controls: this decrease was statistically evident in the GDM group at all three time intervals, while in the control group this decline was not significant between 32 and 36 weeks (Table 3 and Figure 1).

When the same analysis was performed but with the division of GDM patients according to their treatment, a statistically significant difference was noted at the 36-weeks check point. At that week, the NT plus insulin group showed a significant difference in comparison to both the NT (*p* = 0.0113) and control groups (*p* = 0.0289) (Table 4).

The last comparison of UA-PI values was performed with the division of the NT plus insulin group into two subgroups—the long-acting insulin group and the combined therapy group, as shown in Table 5.

The combined insulin group exhibited significantly reduced UA-PI values at 32 and 36 weeks of gestation compared to the NT groups, as shown in Figure 2. The combined insulin group showed always reduced UA-PI values in comparison to the LA insulin group, but the difference never reached statistical significance (Table 5).

## 4. Discussion

A progressive and gradual decline in UA-PI in GDM and control women from the 28th to the 36th week of gestation was observed, as already published [8]. To the best of our knowledge, we here observed for the first time that women who needed insulin therapy had the lowest values of UA-PI at 36 weeks in their AGA fetuses. This difference was not found in the NT group, whose UA-PI values were similar to those of healthy women.

Our results are consistent with the idea that it is clinically more relevant to focus on the level of glycemic control rather than solely on the diabetic condition itself.

Our current finding showing no significant difference between NT women and controls and a significant difference between combined therapy and controls in Doppler indices is consistent with the idea that good metabolic control can protect placental blood vessels from glucose damage. A direct impact of insulin on fetal circulation could theoretically not be excluded, but this is unknown at the moment. Also, due to the reduced number of patients in the combined therapy insulin group, we cannot speculate on the relationship between increasing the dosage of insulin and fetal circulation.

Despite its useful application in relation to several maternal and fetal problems (anemia, hypertensive disorders, and IUGR) [10,11,17,18] fetoplacental Doppler velocimetry still has yet to prove its prognostic efficacy in GDM pregnancies [6,8]. This uncertainty further persists when studying additional fetal vessels, such as the fetal descending aorta and the resistance or the peak systolic velocity of the middle cerebral arteries [8,19].

Altered Doppler indices might be better explained by the consequences (both maternal and fetal) arising from inadequate maternal glycemic control [19,20], rather than hyperglycemia per se.

This could explain the direct relationship between UA-PI values and macrosomia fetuses, as this situation is an unfavorable outcome of diabetes [21,22].

A previous study sought to correlate maternal glycemic status and the UA systolic/end-diastolic (S/D) ratio in the third trimester. Significant statistical variations in the ratio of UA S/D were seen between pregnancies complicated by diabetes that were well-managed and those that were poorly controlled [23]. Another approach to investigating this correlation is based on the evaluation of fetal Doppler indices before and after a 75 g OGTT [9,15].

In low-risk pregnancies at 30–32 weeks of gestation, Haugen et al. found that UA-PI was significantly decreased after testing, with changes related to FHR but independent of fetal size [15]. The same authors, in another study, showed that changes in umbilical vein and fetal liver blood flow were positively correlated with fetal AC, thus suggesting that in LGA fetuses, maternal glucose loading increases blood flow from the placenta to the fetal liver [24].

On a smaller sample of healthy pregnant women, others evaluated the effects of glucose loading after 50-g-OGTT. They observed that UA-RI changed slightly (measured at 36–40 weeks’ gestation) [25], or showed a significant increase when maternal blood glucose was more than 102 mg (at 24–28 weeks’ gestation) [26].

The relationship between OGTT-induced acute hyperglycemia and Doppler velocimetric dynamics changes in UA in healthy pregnant women is not universally confirmed. No correlation was found between UA-PI and maternal age, BMI, parity, z score of AC, HC or EFW in other studies [7,9].

The underlying mechanism of the blood vessel dysfunction seen in GDM might be linked to the L-arginine/nitric oxide (NO) pathway. The well-established features of NO include angiogenic, vasodilatory, and metabolic regulatory effects. NO concentration is a reliable predictor of endothelial homeostasis due to its impact on various aspects of endothelial cell function, including vascular tone, platelet aggregation, and endothelium–leukocyte interaction [27,28]. In in vitro models (human umbilical vein endothelial cells HUVECS), elevated hyperglycemia stimulates L-arginine transport and enhances the release of nitric oxide (NO). Also, insulin promotes the transport of L-arginine, as well as the production of basal NO and PGI2, [29]. This contributes to the difficulty met in discriminating the role of these two stimulatory agents and the endothelial disfunction observed.

The presence and function of nitric oxide synthase (eNOS) and the levels of protein expression of h-CAT-1 were also measured in the umbilical cord blood of individuals with GDM [28].

Unfortunately, besides nitric oxide, there are several additional factors that contribute to endothelial dysfunction in GDM patients, including obesity, chronic low-grade inflammation, insulin resistance, oxidative stress, and altered lipid metabolism. The interactions between these components occur through intricate systems [3,30].

Hence, it is challenging to ascertain the specific component accountable for the vascular alterations observed in our investigation.

There are limitations in our study. First, it is a retrospective study. Secondly, the sample size of the combined insulin therapy group does not allow further subdivision based on the effects of different insulin dosages and fetal circulations.

Moreover, we acknowledge that one of the major limitations of our study is the absence of propensity score-based methods to adjust for differences in baseline clinical characteristics between the treated and not-treated groups. As a result, the observed differences in the PI could be partially or fully attributable to pre-treatment characteristics, rather than the effect of the insulin itself. Despite this, it must be recognized that the two compared groups could not be full homogeneous due to additional risk factors that affect the patient in need of therapy. For all that, this imbalance could introduce a risk of confounding, which may bias our results. Without matching or weighting, our analysis remains vulnerable to confounding by indication. Patients who received the insulin showed different baseline risk profiles compared to those who did not, and these differences could affect both the decision to treat and the clinical outcomes. Although we adjusted for some confounders between the diabetic group and the controls, residual confounding may still occur due to unmeasured or inadequately adjusted variables. Moreover, given the lack of matching, selection bias may have influenced our results. Despite this, to the best of our knowledge, there should be not a difference in the PI in such groups in relation to, for example, the ethnicity. Anyway, patients who were treated with the drug might systematically differ from those who were not in ways that we did not or could not measure (apart from, e.g., ethnicity, BMI, GW, family history of DM, previous macrosomia and/or GDM, fasting blood glucose and HbA1c), leading to biased estimates of treatment effect. This should drive future studies on GDM and umbilical PI.

Our study also has strengths. Here, Doppler ultrasonography parameters were measured only by one expert sonographer; we deliberately did not include infants who were born prematurely, had growth restrictions, or were larger than expected for their gestational age; we measured and analyzed the UA-PI of fetuses with diabetes at three distinct time points in the third trimester of pregnancy; we conducted—for the first time in the literature—separate analyses of the different types of insulin therapy (long-acting versus combination therapy); the study was performed in a single facility where GDM screening and glycemic goals were consistent during the whole study duration.

## 5. Conclusions

A decreased impedance to blood flow in the umbilical artery of diabetic mothers on insulin therapy was observed. This reduction was particularly pronounced during the final stage of pregnancy. Ultrasonography is a well-established approach for the monitoring of fetal–maternal well-being. Nevertheless, the extent to which umbilical artery PI can predict unfavorable outcomes associated with maternal hyperglycemia has yet to be determined.

GDM is a complex syndrome that is linked to negative effects in both the short and long term. The well-established phenomenon of endothelial dysfunction is closely linked to this illness, and other variables (such as chronic low-grade inflammation, oxidative stress, insulin resistance, and altered lipid metabolism) that interact with one another can affect the blood vessels of both the mother and the fetus. Maternal hyperglycemia may exacerbate the condition of the fetal blood vessels.

Further additional prospective studies with a significant number of participants are necessary to confirm the precise impact of glucose levels and medical interventions on the circulations of both the fetus and the mother.

## Figures and Tables

**Figure 1 healthcare-12-01972-f001:**
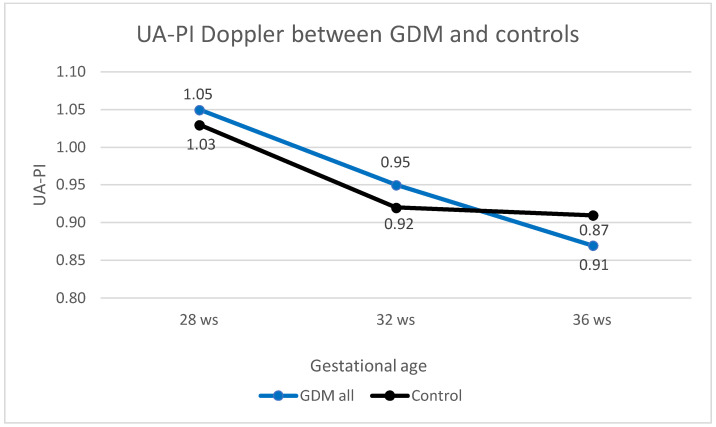
UA-PI trend in GDM and control groups. UA-PI trend during the third trimester of pregnancy, from 28 to 36 weeks, in GDM (blue line) and control groups (black line).

**Figure 2 healthcare-12-01972-f002:**
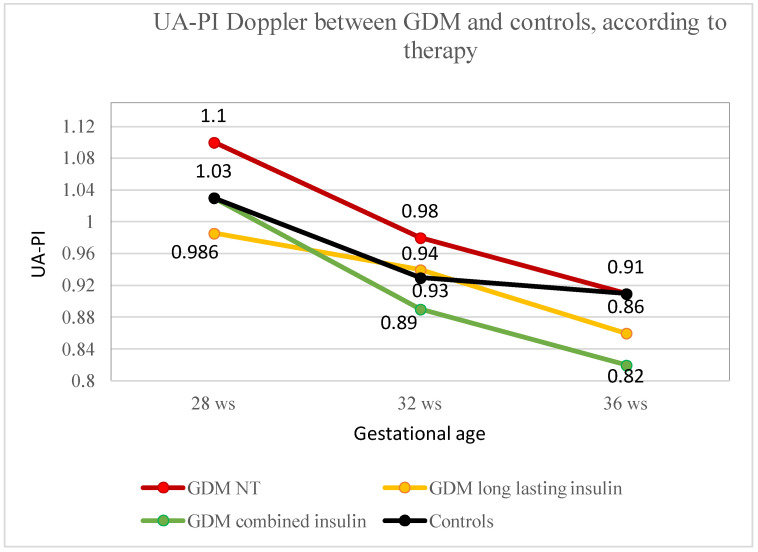
UA-PI trend in NT, long-acting insulin, combined insulin and control groups. UA-PI trend during the third trimester of pregnancy, from 28 to 36 weeks, in NT (red line), long-acting insulin (yellow line), combined insulin (green line) and control groups (black line).

**Table 1 healthcare-12-01972-t001:** Demographic, anamnestic and obstetrical characteristics of study groups.

	NT Group	NT + Insulin Group	*p*-Value
	n = 287	n = 160	
Age (years)	33.32 ± 5.31	34.01 ± 5.22	0.9331
Smoking	15 (5.2)	9 (5.6)	0.8305
Caucasian ethnicity *	217 (75.6)	89 (55.6)	**0.0001**
Parity			**0.0078**
- Nulliparous	120 (41.8)	46 (28.7)	
- Multiparous	167 (58.2)	114 (71.3)	
Pregestational BMI (Kg/m^2^)	25.46 ± 5.33	27.78 ± 5.04	**0.0001**
BMI at delivery	28.61 ± 5.04	30.36 ± 4.63	**0.0003**
GWG (Kg)	8.44 ± 5.39	7.21 ± 5.60	**0.0233**
Onset of pregnancy			0.8457
- Spontaneous	267 (93)	150 (93.7)	
- IVF	20 (7)	10 (6.3)	
Family history of diabetes	114 (39.7)	81 (50.6)	**0.0288**
Previous macrosomia	7 (2.4)	16 (10)	**0.0012**
Previous GDM	33 (11.5)	40 (25)	**0.0003**
Fasting blood glucose I trimester (mg/dL)	83.10 ± 10.55	95.16 ± 20.47	**0.0001**
OGTT			
- Fasting blood glucose (mg/dL)	89.39 ± 9.19	103.10 ± 71.88	**0.0015**
- Blood glucose at 60’ (mg/dL)	165.07 ±30.95	173.24 ±30.86	**0.0076**
- Blood glucose at 120’ (mg/dL)	138.13 ± 28.21	144.80 ± 34.59	**0.0278**
Glycated hemoglobin, %	5.18 ± 0.31	5.54 ± 0.48	**0.0001**

Data are expressed as mean ± standard deviation or as absolute number (percentage). *p* < 0.05 was considered statistically significant (marked in bold). BMI: body mass index. GDM: gestational diabetes mellitus. GWG: gestational weight gain. IVF: in vitro fertilization. OGTT: oral glucose tolerance test. NT: nutritional therapy. * Among the non-Caucasian ethnicities, we recruited 60% black and 40% Asian.

**Table 2 healthcare-12-01972-t002:** Delivery and neonatal outcomes.

	NT Group	NT + Insulin Group	*p*-Value
	n = 287	n = 160	
GA at delivery (weeks)	38.64 ± 1.78	38.34 ±1.71	0.0774
Hypertensive disorders	12 (4.2)	10 (6.2)	0.3653
Induction of labor	74 (25.8)	63 (39.4)	**0.0038**
PROM	92 (32)	47 (29.4)	0.5949
Positive rectal–vaginal swab	25 (8.7)	22 (13.7)	0.1085
Stained amniotic fluid	36 (12.5)	12 (7.5)	0.1121
Mode of delivery			
- VD	207 (72.1)	107 (66.9)	0.2806
- CS	80 (27.9)	53 (33.1)	
PPH	30 (10.4)	29 (18.1)	0.8736
Fetal sex			
- M	151 (52.6)	88 (55)	0.6925
- F	136 (47.4)	72 (45)	
Birthweight (g)	3148.84 ± 533.30	3248.93 ± 501.87	0.0527
APGAR score			
- 1st minute	8.32 ± 1.49	8.24 ± 1.74	0.5799
- 5th minute	8.90 ± 0.68	8.81 ± 1.02	0.2344
APGAR score < 7			
- 1st minute	25 (8.7)	16 (10)	0.7328
- 5th minute	4 (1.4)	3 (1.9)	0.7047
NICU admission	28 (9.6)	13 (8.1)	0.6123

Data are expressed as mean ± standard deviation or as absolute number (percentage). *p* < 0.05 was considered statistically significant (marked in bold). BMI: body mass index. CD: cesarean delivery. F: female. g: grams. GA: gestational age. M: male. NT: nutritional therapy. PPH: postpartum hemorrhage. PROM: premature rupture of membranes. VD: vaginal delivery.

**Table 3 healthcare-12-01972-t003:** UA-PI measurements during the third trimester of pregnancy among all patients with gestational diabetes mellitus and controls (matched for age and BMI).

	GDM All Women	Control Group	*p*-Value
	n = 447	n = 100	
UA-PI at 28 ws	1.06 ± 0.26 (a)	1.03 ± 0.30 (d)	0.3725
UA-PI at 32 ws	0.96 ± 0.30 (b)	0.93 ± 0.27 (e)	0.4153
UA-PI at 36 ws	0.88 ± 0.27 (c)	0.91 ± 0.24 (f)	0.2562
*p*-value	**(a) vs. (b) < 0.0001**	**(d) vs. (e) = 0.0117**	
	**(b) vs. (c) < 0.0001**	**(e) vs. (f) = 0.6098**	
	**(a) vs. (c) < 0.0001**	**(d) vs. (f) = 0.0017**	

Data are expressed as mean ± standard deviation or as an absolute number (percentage). *p* < 0.05 was considered statistically significant (marked in bold). a: UA-PI at 28 ws in GDM group; b: UA-PI at 32 ws in GDM group; c: UA-PI at 36 ws in GDM group; d: UA-PI at 28 ws in control group; e: UA-PI at 32 ws in control group; f: UA-PI at 36 ws in control group.

**Table 4 healthcare-12-01972-t004:** UA-PI measurements during the third trimester of pregnancy among the GDM group (divided according to therapy—NT only versus NT plus insulin) and controls.

	NT Group	NT + Insulin Group	Control Group	*p*-Value
	n = 287	n = 160	n = 100	
**UA-PI at** **28 ws**	1.10 ± 0.24 (a)	1.01 ± 0.29 (d)	1.03 ± 0.30 (g)	**(a) vs. (d) 0.0001**
				**(a) vs. (g) 0.0131**
				(d) vs. (g) 0.4958
**UA-PI at** **32 ws**	0.98 ± 0.31 (b)	0.92 ± 0.29 (e)	0.93 ± 0.27 (h)	(b) vs. (e) 0.0629
				(b) vs. (h) 0.1784
				(d) vs. (h) 0.8986
**UA-PI at** **36 ws**	0.91 ±0.27 (c)	0.84 ± 0.26 (f)	0.91 ± 0.24 (i)	**(c) vs. (f) 0.0113**
				(c) vs. (i) 0.9060
				**(f) vs. (i) 0.0289**
*p*-value	**(a) vs. (b) < 0.0001**	**(d) vs. (e) = 0.0108**	**(g) vs. (h) = 0.0117**	
	**(b) vs. (c) = 0.0028**	**(e) vs. (f) = 0.0062**	(h) vs. (i) = 0.6098	
	**(a) vs. (c) < 0.0001**	**(d) vs. (f) < 0.0001**	**(g) vs. (i) = 0.0017**	

Data are expressed as mean ± standard deviation or as an absolute number (percentage). *p* < 0.05 was considered statistically significant (marked in bold). NT: nutritional therapy. UA-PI: umbilical artery pulsatility index. a: UA-PI at 28 ws in NT group; b: UA-PI at 32 ws in NT group; c: UA-PI at 36 ws in NT group; d: UA-PI at 28 ws in NT + Insulin group; e: UA-PI at 32 ws in NT + Insulin group; f: UA-PI at 36 ws in NT + Insulin group; g: UA-PI at 28 ws in control group; h: UA-PI at 32 ws in control group; i: UA-PI at 36 ws in control group.

**Table 5 healthcare-12-01972-t005:** UA-PI measurements during the third trimester of pregnancy among the GDM insulin group (divided according to insulin therapy—long-acting insulin versus combined insulin) and NT group and controls.

	NT Group	LA Group	Combined Therapy Group	Control Group	*p*-Value
	n = 287	n = 97	n = 63	n = 100	
**UA-PI at** **28 ws**	1.10 ± 0.24 (a)	0.99 ± 0.30 (d)	1.03 ± 0.26 (g)	1.03 ± 0.30 (l)	**(a) vs. (d) = 0.0001**
					(d) vs. (g) = 0.3017
					**(a) vs. (g) = 0.0379**
					**(a) vs. (l) = 0.0131**
					(d) vs. (l) = 0.2939
					(g) vs. (l) = 0.9409
**UA-PI at** **32 ws**	0.98 ± 0.31 (b)	0.95 ± 0.27 (e)	0.89 ± 0.31 (h)	0.93 ± 0.27 (m)	(b) vs. (e) = 0.3276
					(e) vs. (h) = 0.2523
					**(b) vs. (h) = 0.0417**
					(b) vs. (m) = 0.0629
					(e) vs. (m) = 0.6747
					(h) vs. (m) = 0.4233
**UA-PI at** **36 ws**	0.91 ±0.27 (c)	0.86 ± 0.23 (f)	0.82 ± 0.30 (i)	0.91 ± 0.24 (n)	(c) vs. (f) = 0.1099
					(f) vs. (i) = 0.3378
					**(c) vs. (i) = 0.0205**
					(c) vs. (n) = 0.9060
					(f) vs. (n) = 0.1137
					**(i) vs. (n) = 0.0292**
*p*-value	**(a) vs. (b)** **<0.0001**	(d) vs. (e) =0.3176	**(g) vs. (h)** **=0.0062**	**(l) vs. (m)** **=0.0117**	
	**(b) vs. (c)** **=0.0028**	**(e) vs. (f)** **=0.0163**	(h) vs. (i) =0.1734	(m) vs. (m) =0.6098	
	**(a) vs. (c)** **<0.0001**	**(d) vs. (f)** **<0.0010**	**(g) vs. (i)** **<0.0001**	**(l) vs. (n)** **=0.0017**	

Data are expressed as mean ± standard deviation or as absolute number (percentage). *p* < 0.05 was considered statistically significant (marked in bold). NT: nutritional therapy. UA-PI: umbilical artery pulsatility index. a: UA-PI at 28 ws in NT group; b: UA-PI at 32 ws in NT group; c: UA-PI at 36 ws in NT group; d: UA-PI at 28 ws in LA group; e: UA-PI at 32 ws in LA group; f: UA-PI at 36 ws in LA group; g: UA-PI at 28 ws in combined therapy group; h: UA-PI at 32 ws in combined therapy group; i: UA-PI at 36 ws in combined therapy group; l: UA-PI at 28 ws in control group; m: UA-PI at 32 ws in control group; n: UA-PI at 36 ws in control group.

## Data Availability

The data presented in this study are available on request from the corresponding author due to privacy.

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
