# Peer review of "Does Insulin Treatment Affect Umbilical Artery Doppler Indices in Pregnancies Complicated by Gestational Diabetes?"

_healthcare, 2024, doi:10.3390/healthcare12191972_

Round 1

Reviewer 1 Report

Comments and Suggestions for Authors

 - In the methods part of the abstract you mentioned only GDM patients, and then a control group appears in the results part. The methods should be described more precisely and in detail in the abstract.

 - Use "neonatal outcomes" instead of "newborn outcomes"

 - The conclusion part of the abstract should be shortened.

 - Line 62 "worse newborn outcomes" - replace with "adverse neonatal outcomes"

 - I recommend adding data on the prevalence of GDM in Italy.

 - I recommend using the following subsections in the “Material and methods” part: (a) study design (b) study setting; (c) study population; (d) inclusion and exclusion criteria; (e) outcome measures and methods; (f) statistical analysis.

 - Which long-acting and fast-acting insulin analogues were administered?

 - Use "g" instead of "gr"

 - You should use table templates provided by the journal.

 - Figure 2 is unclear. Some of the values ​​cannot be seen.

Author Response

Reviewer 1

Q - In the methods part of the abstract you mentioned only GDM patients, and then a control group appears in the results part. The methods should be described more precisely and in detail in the abstract.

A – We thank the reviewer for noticing this and we agreed the healthy control group should be mentioned in the abstract. We provided to add the 100 healthy pregnancies taken as control group in the abstract.

Q - Use "neonatal outcomes" instead of "newborn outcomes"

A – We provided to substitute “newborn” with “neonatal”. It sounds more appropriate indeed. Thanks.

Q - The conclusion part of the abstract should be shortened.

A – We thank the reviewer for this comment. We provided to shorten the abstract’s conclusion and made it more clear ad easy to read.

Q - Line 62 "worse newborn outcomes" - replace with "adverse neonatal outcomes"

A - We provided to substitute “newborn” with “neonatal”. It sounds more appropriate indeed. Thanks.

Q - I recommend adding data on the prevalence of GDM in Italy.

A – We agree indeed with the reviewer that this is an important data. We provided to add the GDM prevalence in our country. It is around 10.9%. Thanks

Q - I recommend using the following subsections in the “Material and methods” part: (a) study design (b) study setting; (c) study population; (d) inclusion and exclusion criteria; (e) outcome measures and methods; (f) statistical analysis.

A – We provided to divide the methods according to the reviewer suggestions. The methods section is now more appropriate indeed. Thank you.

A - Which long-acting and fast-acting insulin analogues were administered?

Q – As long-acting it was used the Glargine, as fast-acting the Lispro. This has been now specified in the text. Many thanks for pointing this out.

Q - Use "g" instead of "gr"

A- We provided to substitute “g” instead of “gr” to indicate grams, as per reviewer’s advice. Thanks.

Q - You should use table templates provided by the journal.

A – All the tables have been revised according to the journal template. Many thanks for pointing this out.

 Q - Figure 2 is unclear. Some of the values ​​cannot be seen.

A – Figure 2 has been modified. Now all the values can be red. Thanks

Reviewer 2 Report

Comments and Suggestions for Authors

Dear authors

GDM itself is pathology of pregnancy, whose percentage is in increasing. Untill nowm there was no research on influence of therapy of GDM on PI AU and fetal CD.This is very good research, and conclusions are relevant.

Thanks to yours  research, there is basis for another researches with more participants to confirm or decline yr thesis.

Research itself is very well conducted and quality of methodology is good.

I have just really minor coments, more thehnical.

In line 80...according to what society guidlines have you decided to take oGTT in 16th and then 24th week of gestation? Input reference.

In line 125, also imput reference.

Also , in mateials and method, you should input that research is made in accordance with Helsinki declaration, and input reference

All other in paper seems fine.

Regards, and good luck

Comments on the Quality of English Language

English language is fine.

Author Response

Reviewer 2

Q - GDM itself is pathology of pregnancy, whose percentage is in increasing. Untill nowm there was no research on influence of therapy of GDM on PI AU and fetal CD.This is very good research, and conclusions are relevant. Thanks to yours  research, there is basis for another researches with more participants to confirm or decline yr thesis. Research itself is very well conducted and quality of methodology is good.

I have just really minor coments, more thehnical.

A – We really appreciated the positive comments of the reviewer. We would like to thank them for their time.

Q - In line 80...according to what society guidlines have you decided to take oGTT in 16th and then 24th week of gestation? Input reference.

A – The guidelines we refer to are the national Italians. We provided to specify and pointed out the reference.

Q – In line 125, also imput reference.

A – We provided to add the appropriate link to check out the FMF certification as pointed out by the reviewer. Thanks.

Q - Also, in mateials and method, you should input that research is made in accordance with Helsinki declaration, and input reference

A – This has been added. Thanks a lot.

Q - All other in paper seems fine.

A – Thanks again for your time and positive comments.

Reviewer 3 Report

Comments and Suggestions for Authors

Comments/suggestions:

1.     Introduction

§  Suggest explaining the rational of assessing umbilical artery doppler indices (UA-PI) in patients with GDM, as it is a marker of placental insufficiency and consequent intrauterine growth restriction or suspected pre-eclampsia. However, the most common adverse outcome for GDM is LGA. UA-PI may not be a good marker for LGA.

2.     Materials and Methods          

§  Could you explain why pregnancies with coexisting conditions were excluded?

                                               i.     For example, LGA is often considered an adverse outcome of GDM. By excluding pregnancies with LGA – they results may be biased (i.e., pregnancies with worst outcomes were not included in the study)

§  Multiple comparisons should be adjusted

§  Although age and BMI were matched for GDM and controls, other potential confounders, such as parity and race/ethnicity, should be adjusted when making comparisons.

3.     Results

§  Table 1

                                               i.     Could you describe what ethnicities were the non-Caucasian women? For examples, mostly blacks?

§  Table 2

                                               i.     LGA and SGA should be added

§  Table 3 & Figure 1 as well as Table 4 and Figure 2 reported redundant information – suggest condensing all information in figures.

Author Response

Reviewer 3

Introduction

Q-  Suggest explaining the rational of assessing umbilical artery doppler indices (UA-PI) in patients with GDM, as it is a marker of placental insufficiency and consequent intrauterine growth restriction or suspected pre-eclampsia. However, the most common adverse outcome for GDM is LGA. UA-PI may not be a good marker for LGA.

A – We thank the reviewer for pointing this out. Although the UA PI is not a marker for GDM, we know that it is one of the best markers for the fetal well-being. It is to be considered also how easy and immediate is to assess the UA PI, and how useful this could be also for non-experienced obstetricians and fetal medicine doctors. On the other hand, we know that GDM affect the fetal well-being and in particular the maternal-fetal circulation, inducing endothelial dysfunction. It is also interesting to study and understand whether the mother's poor glycemic control, resulting in hyperglycemia, affects the fetal circulation.

Based on this and given that not a lot of studies on this have been conducted, we decided to assess UA PI and GDM, with a focus on the various subgroup of women undergoing different treatments. We provided to specify this at the end of the introduction. Thanks again for noticing this.

Materials and Methods         

Q - Could you explain why pregnancies with coexisting conditions were excluded?

For example, LGA is often considered an adverse outcome of GDM. By excluding pregnancies with LGA – they results may be biased (i.e., pregnancies with worst outcomes were not included in the study).

A – This is actually a very interesting comment that helped us clarifying this point that was otherwise tricky. We provided to exclude other pregnancy complication to avoid being biased in the measurement of UA PI, but to be as much as possible clear in the measurement in the case of GDM (we wanted only to stratify the UA PI according to the GDM therapy that the patient is following). We ruled out maternal pathologies preexisting to pregnancy that could in any way alter maternal fetal circulation (e.g., hypertensive disorders).

Regarding LGA babies, it is true that we excluded them to avoid being affected by LGAs themselves, in agreement with previous studies that established that fetal macrosomia is associated with lower AO-PI values.

  • Sirico A, Rizzo G, Maruotti GM et al (2016) Does fetal macrosomia affect umbilical artery Doppler velocity waveforms in pregnancies complicated by gestational diabetes? J Matern Fetal Neonatal Med 29(20):3266-70. doi: 10.3109/14767058.2015.1121479. Epub 2015 Dec 23.
  • Maruotti GM, Rizzo G, Sirico A, et al. Are there any relationships between umbilical artery Pulsatility Index and macrosomia in fetuses of type I diabetic mothers? J Matern Fetal Neonatal Med 2014;27:1776–81

Q - Multiple comparisons should be adjusted. Although age and BMI were matched for GDM and controls, other potential confounders, such as parity and race/ethnicity, should be adjusted when making comparisons.

A – While we agree with the reviewer that potential confounding factors may exist concerning the disparities between the two examined groups, which might introduce bias, we assert that correction is not feasible, at least in this first investigation, owing to the study's inherent character. This research is the first analysis of the disparities in PI between treated and untreated GDM patients, with particular emphasis on the various types of medication. We assert that patients exhibit inherent variations, such as the important one shown in Table 1, which may influence the need of treatment, particularly in the context of GDM. We also contend that more prospective research are necessary for a comprehensive analysis of this matter. In accordance with the reviewer's observations, we conducted a thorough analysis of this issue, addressing and explaining this possible prejudice within our limitations. We express our gratitude to the reviewer for recognizing this and enabling us to disclose it, so assisting future researchers in this area. 

Results

Q -  Table 1

Could you describe what ethnicities were the non-Caucasian women? For examples, mostly blacks?

A – We thank the reviewer for pointing out this important aspect. Among different ethnicity than Caucasian we recruited 60% of Black patient and 40% of Asian. This has been specified in the table 1. Thanks again

Q -  Table 2

LGA and SGA should be added

A – We thank the reviewer for pointing this out. We preferred to report the birthweight since the table is summarizing neonatal outcomes and not fetal characteristics. Moreover, the birthweight is itself a more precise value than the estimate fetal weight. Thanks again.

Q -  Table 3 & Figure 1 as well as Table 4 and Figure 2 reported redundant information – suggest condensing all information in figures.

A – We thank the reviewer for this comment, but we would like to leave both table (summarizing results) and figures (making the results easy and immediate to read) for the sack of completeness. Thanks again.

Round 2

Reviewer 1 Report

Comments and Suggestions for Authors

I am satisfied with the author’s responses to my questions/issues raised in my initial review.